# Plasma Extracellular Vesicle Characteristics as Biomarkers of Resectability and Radicality of Surgical Resection in Pancreatic Cancer—A Prospective Cohort Study

**DOI:** 10.3390/cancers15030605

**Published:** 2023-01-18

**Authors:** David Badovinac, Katja Goričar, Teja Lavrin, Hana Zavrtanik, Vita Dolžan, Metka Lenassi, Aleš Tomažič

**Affiliations:** 1Department of Abdominal Surgery, University Medical Centre Ljubljana, Zaloška 7, 1000 Ljubljana, Slovenia; 2Department of Surgery, Faculty of Medicine, University of Ljubljana, Vrazov trg 2, 1000 Ljubljana, Slovenia; 3Institute of Biochemistry and Molecular Genetics, Faculty of Medicine, University of Ljubljana, Vrazov trg 2, 1000 Ljubljana, Slovenia

**Keywords:** liquid biopsy, biomarkers, extracellular vesicles, pancreatic ductal adenocarcinoma, resection, patient stratification

## Abstract

**Simple Summary:**

Despite diagnostic workup, unresectable or metastatic disease is still often found in PDAC patients at surgery, leading to unnecessary laparotomy and delay in systemic treatment. Liquid biopsy with sEV from plasma provides a direct insight into tumor characteristics and biology and has been shown to be valuable for diagnosis and treatment surveillance in different types of cancer. In patients with PDAC, deemed resectable upon diagnostic workup, preoperative concentrations of plasma sEV differ between patients who will undergo tumor resection and those who will solely have exploration without resection. Furthermore, among patients with resection, preoperative sEV concentrations differ between patients who will undergo radical (R0) resection and those with microscopic or macroscopic tumor remnant. In the future, liquid biopsy with sEV concentrations could provide important complementary information for better stratification of patients with presumably resectable PDAC and could assist in the decision to postpone surgery for neoadjuvant therapy or avoid surgery with possible complications altogether.

**Abstract:**

Due to possible diagnostic misjudgment of tumor resectability, patients with pancreatic ductal adenocarcinoma (PDAC) might be exposed to non-radical resection or unnecessary laparotomy. With small extracellular vesicles (sEV) obtained by liquid biopsy, we aimed to evaluate their potential as biomarkers of tumor resectability, radicality of resection and overall survival (OS). Our prospective study included 83 PDAC patients undergoing surgery with curative intent followed-up longitudinally. sEV were isolated from plasma, and their concentration and size were determined. Fifty patients underwent PDAC resection, and thirty-three had no resection. Preoperatively, patients undergoing resection had higher sEV concentrations than those without resection (*p* = 0.023). Resection was predicted at the cutoff value of 1.88 × 10^9^/mL for preoperative sEV concentration (*p* = 0.023) and the cutoff value of 194.8 nm for preoperative mean diameter (*p* = 0.057). Furthermore, patients with R0 resection demonstrated higher preoperative plasma sEV concentrations than patients with R1/R2 resection (*p* = 0.014). If sEV concentration was above 1.88 × 10^9^/mL or if the mean diameter was below 194.8 nm, patients had significantly longer OS (*p* = 0.018 and *p* = 0.030, respectively). Our proof-of-principle study identified preoperative sEV characteristics as putative biomarkers of feasibility and radicality of PDAC resection that also enable discrimination of patients with worse OS. Liquid biopsy with sEV could aid in PDAC patient stratification and treatment optimization in the future.

## 1. Introduction

With dismal prognosis and a five-year survival rate below 10%, pancreatic cancer ranks among the deadliest malignancies. It already represents the fourth leading cause of cancer-related deaths worldwide and its incidence in developed countries is rising even further [1,2,3]. Pancreatic ductal adenocarcinoma (PDAC) is the most common histologic type of pancreatic cancer, representing more than 90% of all cases. Only about 20% of PDAC patients are eligible for surgical resection at the time of diagnosis, with the rest having locally advanced or metastatic disease, where resection is no longer indicated [3,4]. Upfront surgical resection is still a cornerstone of management of resectable pancreatic cancer, with adjuvant therapy applied in order to improve long-term survival. In borderline resectable PDAC, achieving radical (R0) resection is not likely, and neoadjuvant systemic therapy is often initiated with the aim of downsizing the tumor and making it resectable. Unfortunately, even modern radiological workup sometimes misjudges the actual resectability status of PDAC, as up to 23% of presumably resectable tumors are found unresectable or metastatic at laparotomy [5]. Furthermore, little information on tumor biology can be obtained with standard diagnostic tools before operation. Routinely acquired serum carbohydrate antigen 19-9 (CA 19-9) levels and tumor differentiation, determined by fine needle aspiration, have limitations and often fail to provide reliable information about PDAC characteristics [6,7,8]. However, a more precise evaluation of the tumor could aid in better stratification of potential candidates for surgery and a more tailored approach to their treatment.

Novel diagnostic approaches, such as liquid biopsy, may provide additional preoperative data and help stratify patients according to the potential of achieving resection of PDAC. With an aim of diagnosing and monitoring different diseases, including cancers, liquid biopsy samples body fluids (e.g., blood) and provides material for specific analyses. Liquid biopsy for PDAC is under-researched, yet, to some extent, it can be used as a substitute for tissue biopsies [9,10]. PDAC-derived material from peripheral blood, such as circulating tumor cells, circulating tumor DNA and extracellular vesicles (EV), provide insight into genetic alterations and tumor biology or help with disease diagnosis and response to treatment [10,11,12]. Such information could be used complementary to preoperative radiological imaging in order to determine PDAC resectability more precisely, help evaluate radicality of tumor resection and follow the patient’s response to surgical or systemic treatment. Liquid biopsy could thus aid in patient stratification and identify those who would benefit most from immediate chemotherapy, even in presumably resectable disease, and those who could avoid futile surgical treatment with potential postoperative complications in aggressive or advanced PDAC.

EV are a liquid biopsy biomarker that has been gaining in popularity in the field of cancer research. They are a heterogeneous group of membrane-bound particles subdivided into exosomes, microvesicles and apoptotic bodies according to their size and site of formation. EV derive from all types of body cells and can thus be found in all body fluids, including pancreatic juice and blood [13,14]. Furthermore, biophysical characteristics and molecular content of EV (e.g., nucleic acids and proteins) are a direct reflection of the physiological and pathological state of the cell of origin, including tumor cells. EV cargo is protected from degradation in circulation [15]. Since EV originate from living cells, they can be detected at an early disease stage [15,16,17]. This is an important advantage to other circulating biomarkers, which are released in circulation after cell necrosis or apoptosis, a phenomenon associated with advanced tumor stage. All of these EV traits offer diagnostic and therapeutic opportunities for EV utilization, including in cancer [11,14].

In PDAC, EV are an important factor in tumor pathogenesis, prevarication of the immune system, intercellular communication and local or metastatic progression of the disease [13]. Biological material such as different proteins and nucleic acids from within EV has been shown to correlate with PDAC survival and stage [18,19,20,21,22,23,24,25]. Even biophysical characteristics of EV correlate with biological features of PDAC. Poorly differentiated PDAC is associated with larger plasma EV compared to well/moderately differentiated tumors, and the same has been demonstrated in metastatic disease, where EV were larger than in localized PDAC [11,22]. Furthermore, EV concentrations and their cargo were shown to be associated with early detection of different cancers, their progression and response to treatment [11,14,17,26]. The concentration of EV and their protein levels were associated with tumor differentiation in glioma and colorectal cancer, while EV size and concentrations were predictive of disease-free survival and overall survival (OS) in lung, prostate, colorectal and esophageal cancers [27,28,29,30,31,32]. Such features make EV interesting biomarkers for potential stratification of cancer patients, including in PDAC.

The goal of our proof-of-principle study was to evaluate the potential of small plasma EV (sEV) as prognostic factors of tumor resectability and OS; moreover, we wanted to evaluate their application in grading radicality of tumor resection. Based on these associations, we aimed to potentially stratify PDAC patients into prognostic groups both preoperatively and postoperatively. To achieve this, a prospective cohort of patients with PDAC undergoing surgery with curative intent was enrolled, and association of plasma sEV characteristics with OS, PDAC resectability and radicality of surgery was evaluated. Longitudinal follow-up was scheduled in order to evaluate the dynamics of EV characteristics with regard to treatment; thus, timed blood samples were collected for up to one year after surgery.

## 2. Materials and Methods

### 2.1. Study Design and Data Collection

This prospective cohort study included patients with preoperatively confirmed or suspected diagnosis of PDAC. All subjects were presumed to have a resectable tumor based on diagnostic workup and were presented at a multidisciplinary team meeting that also indicated surgery; none were suspected to have distant metastases or locally unresectable disease before the operation. The enrolled patients underwent surgery with curative intent at the Department of Abdominal Surgery, University Medical Centre Ljubljana, Ljubljana, Slovenia, in the period from 1 January 2018 to 31 December 2019. Depending on the intraoperative assessment of the extent of the disease, patients underwent either surgical resection or exploration without resection. If diagnosis of PDAC was refuted by histopathological examination of the resected tissue or by intraoperatively obtained tissue biopsy in case of sole surgical exploration, patients were excluded from the study. Neoadjuvant therapy was considered an exclusion criterion. The study was approved by the Republic of Slovenia National Medical Ethics Committee (Study No. 0120-155/2016-2, KME 106/03/16) and conducted in accordance with the Declaration of Helsinki. Prior to enrolment, written informed consent was obtained from all subjects.

Patient data were collected before and during surgery and again one, six and twelve months postoperatively. These included patient demographic data (including sex, age and weight), American Society of Anesthesiologists (ASA) score, body mass index (BMI), alcohol consumption, smoking status, tumor size on preoperative computed tomography scan and adjuvant chemotherapy (if applicable). Laboratory blood test analysis included white blood cell (WBC) count, C-reactive protein (CRP), CA 19-9 and carcinoembryonic antigen (CEA). The pathology report of the resected or biopsied tissue included surgical resection status (margin-negative or R0 resection, microscopically positive margins or R1 and macroscopically positive margins or R2), tumor differentiation (well, moderate or poor) and tumor TNM classification [33]. Any missing patient data due to follow-up non-attendance (poor health, disease progression and death) are clearly indicated. Patients’ vital status was determined on 1 January 2021.

Immediately before surgery, blood samples for EV isolation were collected in K2-EDTA collection tubes (6 mL). Blood was again collected during the follow-up one, six and twelve months postoperatively. All the samples were processed by centrifugation at 2500× *g* for 10 min at 4 °C within 4 h after collection, and plasma aliquots were stored at −80 °C. If visually positive hemolysis was present, the samples were excluded from further analysis; thus, at each analysis, the number of patients included is clearly indicated.

### 2.2. Small EV Isolation from Blood Plasma

One milliliter of plasma was thawed on ice and centrifuged at 10,000× *g* for 20 min at 4 °C. The supernatant was diluted with phosphate-buffered saline (PBS) to 9 mL and pipetted over 2 mL of 20% sucrose in 13 mL tubes. After 2 h 15 min centrifugation at 100,000× *g* (at 4 °C) (MLA-55 in Optima MAX-XP, Beckman Coulter, Brea, CA, USA), supernatant was aspirated, the pellet was suspended in 60 μL of PBS and aliquots were stored at −20 °C until analysis. The described EV isolation protocol was extensively studied in Holcar et al. [34], uploaded into EV-TRACK knowledgebase (EV-TRACK ID: EV200196) [35] and used in several published EV biomarker studies [11,36,37].

### 2.3. Quantification of sEV Concentration and Size

Nanoparticle-tracking (NTA) analysis with the NanoSight NS300 instrument (488 nm laser) connected to an automated sample assistant (both Malvern Panalytical, Malvern, UK) was used to determine sEV size and concentration. Samples were diluted 200 and 400 times in particle-free PBS to reach an optimum concentration range of 1 × 10^8^–10^9^ particles/mL (10–100 particles per frame (PPF)). Five 60 s movies per sample at camera level 14 were recorded and manually examined. In the event of significant abnormalities, up to two (from five) videos per sample were eliminated. Measurements were performed in duplicate. Raw data were analyzed by the NanoSight NTA 3.3 program at the following settings: water viscosity, temperature of 25 °C, detection threshold of 5, minimum track length of 10 and default minimum expected particle size and blur settings [11]. Median (25–75%) PPF in study patients before surgery and one, six or twelve months after surgery was 20.48 (15.29–26.93), 23.75 (16.45–33.35), 24.70 (19.35–32.65) and 21.70 (17.65–31.15), respectively (Appendix A). Output data were expressed as sEV size (the mean hydrodynamic diameter in nm) and concentration (number of particles per 1 mL plasma). Such a quantification approach is repeatable and reliable, and it shows potential for its clinical utilization [11,34].

High-density lipoproteins (HDL) can co-isolate with EV; therefore, we performed ELISA specific for apolipoprotein A1 (ApoA1) in all EV samples (ApoA1: #3710-1HP-2, Mabtech, Nacka Strand, Sweden) as described previously [33]. Measured ApoA1 concentrations were normalized to 1 mL of starting plasma, and any potential dilution of the samples was accounted for. EV concentration did not correlate with ApoA1 concentration at any of the time points (all *p* > 0.05).

### 2.4. Statistical Analysis

All analyses were performed using IBM SPSS Statistics, version 27.0 (IBM Corporation, Armonk, NY, USA). The median and 25–75% range or frequencies were used to describe continuous and categorical variables, respectively. The Mann–Whitney test and Fisher’s exact test were used to compare the distribution of continuous variables and categorical variables between different patient groups, respectively. For comparison of continuous variables in different time points, the Wilcoxon signed-rank test for related samples was used. Only patients with data available for all time points evaluated in a comparison were included, and the number of patients included in each analysis is clearly indicated. Spearman’s rho correlation coefficient (ρ) was used to evaluate correlations between continuous variables. A receiver operating characteristic (ROC) curve was used to determine the area under the curve (AUC) and cutoff with the highest sum of specificity and sensitivity. In survival analysis, OS was defined as the time from surgery to death from any cause. Kaplan–Meier analysis was used to calculate the median OS and follow-up times. Univariable and multivariable Cox regression was used to calculate hazard ratios (HRs) and their corresponding 95% confidence intervals (CIs). Clinical variables used for adjustment in multivariable Cox regression analysis were selected among all reported clinical variables using stepwise forward conditional selection. All statistical tests were two sided, and the level of significance was set to 0.05.

## 3. Results

### 3.1. Patient Characteristics

Table 1 presents characteristics of all 83 patients enrolled in the study. Tumor resection was performed in 50 patients, while 33 patients underwent exploration without resection. Among patients with resection, 32 had radical tumor resection (8 patients with R0 < 1 mm, 16.3%; 24 with R0 > 1 mm, 49.0%), resection margins were microscopically positive in 15 patients (R1; 30.6%) and 2 had macroscopic residual tumor (R2; 4.1%). For one patient, resection margins were not described. When comparing patients with and without resection (Table 1), BMI six months before surgery was significantly higher in patients without resection (*p* = 0.014). Significant difference was also observed in pT stage (*p* < 0.001) and in presence of distant metastases (*p* < 0.001) as in tumor differentiation, where poor differentiation was more likely to be present in patients without resection (*p* = 0.047).

### 3.2. Patients’ Plasma Small Extracellular Vesicle Characteristics

Concentration and size of plasma sEV were determined for 82 patients immediately before surgery and again one (N = 53, 64.6%), six (N = 43, 52.4%) and twelve months after surgery (N = 29, 35.4%) (Figure 1, Table 2). First, sEV characteristics one month after surgery were compared to preoperative values (Appendix A). In the whole study group, mean diameter was significantly larger one month after surgery in comparison to preoperative values (increased in 32 out of 52 patients, *p* = 0.018). No significant differences in plasma sEV characteristics in later time intervals (past one month after surgery) were found (all *p* > 0.05); however, the number of patients included in these comparisons was smaller due to losing patients during follow-up (Figure 1).

### 3.3. Comparison of Plasma Small Extracellular Vesicle Characteristics between Patients with and without Resection

When comparing patients with and without resection (Figure 1 and Table 2), there was a significant difference in sEV concentration between the two observed groups before surgery; patients undergoing resection of PDAC had higher concentrations of sEV than those without resection (median 2.14 × 10^9^/mL vs. 1.66 × 10^9^/mL, *p* = 0.023, Table 2). Plasma sEV concentrations remained slightly higher in the group of patients with resection during postoperative follow-up (Figure 1), although the difference was no longer statistically significant. No statistically significant differences in sEV size were found between patients with and without resection at any time point, even though patients with resection tended to have smaller EVs before surgery (*p* = 0.057, Table 2).

Using ROC curve analysis, we determined cutoff values for sEV characteristics before surgery to discriminate between patients with resection of PDAC and those without resection. At the cutoff value of 1.88 × 10^9^/mL for preoperative EV concentration, specificity for predicting resection was 0.606, and sensitivity was 0.673 (AUC of 0.649 (95% CI = 0.524–0.774), *p* = 0.023). For the mean diameter before surgery, the cutoff value was 194.8 nm, predicting resection with a specificity of 0.515 and sensitivity of 0.875 (AUC of 0.624 (95% CI = 0.492–0.756), *p* = 0.057).

Additionally, sEV characteristics one month after surgery were compared to preoperative values in each treatment group separately (Appendix A, Figure 1). In the group of 36 patients with resection, a significant rise in mean diameter of sEV was seen in 26 out of 36 patients (*p* = 0.014), while no significant association was seen in 16 patients without resection (*p* = 0.796). On the other hand, no significant changes were observed for sEV concentration within each group (Appendix A).

### 3.4. Association between Radicality of Resection and Plasma sEV Characteristics

The group of patients with resection of PDAC was further divided into two subgroups depending on the radicality of resection (Appendix A, Figure 1). Thirty-one specimens were analyzed from patients who underwent radical (R0) resection, and seventeen were analyzed from patients with micro- or macroscopic tumor remnant (R1 or R2). We observed a significant difference in sEV concentration before surgery between patient subgroups, with patients with R0 resection demonstrating higher plasma sEV concentration than patients with R1 or R2 resection (median 2.68 × 10^9^/mL vs. 1.85 × 10^9^/mL, *p* = 0.014; Figure 2). No significant differences were found with regard to plasma sEV size between the two subgroups pre- and postoperatively.

Furthermore, sEV characteristics one month after surgery were compared to preoperative values in each group separately (Appendix A, Figure 1). In the subgroup of patients with radical (R0) resection, there was a significant rise in mean diameter after one month (in 17 out of 22 cases, *p* = 0.010, Figure 1 right). On the other hand, sEV concentration tended to increase in patients with R1 or R2 resection (in 10 out of 14 cases, *p* = 0.041, Figure 1 left).

### 3.5. Association between Plasma sEV Characteristics and Overall Survival

The median follow-up time for our patient cohort was 25.7 (18.3–28.8) months, with a median overall survival (OS) of 11.3 (5.7–25.3) months. Patients without resection and without systemic chemotherapy demonstrated the poorest median survival of only 3.3 (2.4–8.5) months. Somewhat better was survival in patients without resection who received chemotherapy (8.9 (4.7–12.9) months; HR = 0.47 (0.23–0.94), *p* = 0.034). Survival was further improved when resection was performed; patients with resection and without chemotherapy demonstrated survival of 13.7 (5.3–18.9) months (HR = 0.24 (0.11–0.52), *p* < 0.001), while those with resection and chemotherapy had a survival of 28.3 (13.6–28.3) months (HR = 0.09 (0.04–0.19), *p* < 0.001; Appendix A). Apart from the type of surgery and chemotherapy, higher CA19-9 was also associated with worse survival in the whole group (*p* < 0.001). In a univariable analysis, pT stage four was significantly associated with shorter OS (*p* < 0.001), but it did not remain significant in multivariable analysis.

Association of concentration and size of sEV before surgery with OS was evaluated in the whole study cohort as well as in patients with and without resection separately (Table 3). There were no significant associations with OS in the whole study group or in patients with resection. However, among patients without resection, patients with larger sEV had longer OS in univariable analysis (mean diameter: HR = 0.81, 95% CI = 0.68–0.97, *p* = 0.021). The association was no longer significant after adjustment for significant clinical variables (*p* = 0.065; Table 3).

Due to differences among patients with and without resection, we also evaluated the association with OS if the patients were stratified according to the cutoff values best discriminating between patients with and without resection in the ROC curve analysis (see above) (Table 4). If sEV concentration was above 1.88 × 10^9^/mL, patients had significantly longer OS, with a value of 16.1 (7.7–28.3) months compared to 7.8 (3.8–13.8) months in patients with the concentration below the cutoff value (HR = 0.54, 95% CI = 0.32–0.90, *p* = 0.018; Figure 3A). Additionally, OS was longer (13.7 (6.5–28.3) months vs. 8.5 (3.8–13.8) months) if sEV mean diameter was below 194.8 nm (HR = 1.81, 95% CI = 1.06–3.10, *p* = 0.030; Figure 3B). However, the difference was no longer significant after adjustment for clinical parameters.

## 4. Discussion

This proof-of-principle study is to our knowledge the first to anticipate resectability of PDAC based on sEV concentration and to correlate sEV concentration with radicality of surgical resection. Patients who underwent surgical resection had higher preoperative concentrations of sEV than patients without resection. Moreover, sEV concentration was higher in patients undergoing radical resection when compared to resection with micro- or macroscopically positive margins.

Surgical resection is the only potentially curative treatment option in PDAC, and performing radical resection is the main goal. In resectable disease, the gold standard of treatment remains upfront surgery with curative intent followed by systemic therapy. Negative resection margins (R0), negative lymph nodes (N0) and small tumor size are amongst most important prognostic factors of long-term survival [8,38,39]. However, despite precise diagnostic imaging modalities, only about 20% of patients are eligible for surgery at the time of diagnosis, as surgery offers no survival benefit in locally advanced or metastatic disease [4,40,41,42]. Computed tomography (CT) scan has limited sensitivity for discovering small liver or peritoneal metastatic lesions [43]. Staging laparoscopy is sometimes utilized when advanced disease is suspected to confirm (or rule out) potential metastases, but it offers little help in estimation of local resectability [4,40]. However, as none of the currently available preoperative diagnostic tools estimate the feasibility of radical resection of PDAC accurately enough, liquid biopsy could apply here.

Liquid biopsy has been shown to improve detection of PDAC or distant metastases and to be useful in prediction of tumor recurrence or survival [14,18,44,45,46]. Additionally, in liver, lung, colorectal and breast cancer, liquid biopsy is able to evaluate OS and help monitor disease response to treatment [16,26,31,47,48]. Nonetheless, no studies that would evaluate liquid biopsy predictive value for feasibility of resection of presumably localized PDAC have been conducted. In our study, we have shown that preoperative sEV concentration differs between patients in whom the tumor could have been radically resected and those with margin-positive or unresectable disease. Additionally, analysis of biophysical characteristics of sEV one month after resection revealed a significant enlargement of sEV in patients with resection (larger mean diameter in 72.2% of patients) and, furthermore, an enlargement in the subgroup of patients with R0 resection (larger mean diameter in 77.3%). We hypothesize that sEV released from PDAC before surgery were smaller and that after resection of the tumor, there was a shift towards larger sEV in plasma.

These results seem to be in concordance with our analysis of the association of sEV characteristics before surgery with OS. Patients with a preoperative sEV mean diameter below 194.8 nm had significantly prolonged OS compared to patients above these cutoff values. This analysis was performed on the whole patient cohort, yet better survival is of course expected with tumor resection. Provided with such information of lower likelihood of resection and worse OS, a decision for delaying surgery and initiating neoadjuvant chemotherapy may be more easily supported. Furthermore, unnecessary laparotomy or exploration could thus be avoided together with possible associated postoperative complications. In this manner, liquid biopsy of tumor-derived sEV could be used complementary to standard diagnostic tools for better preoperative characterization of PDAC and a more personalized approach to patients’ treatment, but further studies are needed. Similar use of liquid biopsy has already been established with DNA in some other cancers (ovarian, breast, lung, metastatic colorectal and prostate cancer), and specific tests have even been approved by the U.S. Food and Drug Administration (FDA) [49,50].

In borderline PDAC, neoadjuvant chemotherapy is often initiated with the aim of downstaging the disease to increase the possibility of a microscopically complete (R0) resection [4,40]. All patients at high risk of positive resection margins should be considered as poor candidates for upfront surgery and should be offered neoadjuvant treatment [40], as survival of patients with positive resection margins is comparable to those treated with only chemoradiation (without surgery) [41,42]. However, evaluating resection margins is possible only after thorough histologic examination of the resected specimen, which is often challenging. There is currently still a lack of consensus on acceptable resection margins in PDAC, with some defining R1 as presence of tumor in the margin (at 0 mm) and others defining it as presence of tumor within 1 mm of the margin (<1 mm) [4,40,51]. A standardized evaluation of these margins would indeed result in better stratification of PDAC patients for postoperative systemic therapy after resection; however, there seems to be no significant difference in the outcome after PDAC resection with ≥1 mm or with 0 mm margin clearance [52,53,54,55]. Yet, regardless of the definition of R0 (whether >0 mm or ≥1 mm), additional information about resection radicality would further aid the decision about adjuvant therapy. Based on our proof-of-principle study, plasma concentrations of sEV, acquired by liquid biopsy, could substantiate histopathological findings of negative resection margins or even clarify potential uncertainty in histology after resection. This way, decisions about adjuvant treatment could be easier. Furthermore, in cases of preoperatively assumed high risk of positive resection margins and in radiologically borderline resectable tumors, low plasma sEV concentrations could further confirm high probability of non-radical resection and thus make decisions for neoadjuvant chemotherapy more straightforward. Although neoadjuvant treatment is currently being used mostly in borderline resectable cases, it is becoming increasingly recognizable and established in treatment of even resectable PDAC. This study, along with similar studies in the future, could argue for such an approach in resectable PDAC in order to improve surgical outcomes and survival.

Despite a relatively large number of patients included and sampling plasma in four different time intervals, our proof-of-principle study has some drawbacks. With the natural history and poor prognosis of PDAC, patients were relatively rapidly lost to follow-up, and, consequently, the number of specimens available for analysis in later time intervals decreased. Most of our results one, six and twelve months after surgery were therefore based on smaller study groups, which could have affected statistical significance of performed analyses. Furthermore, the majority of patients after surgery, whether with their PDAC removed or not, were receiving adjuvant therapy one month later, and that could interfere with biophysical characteristics of sEV. For a similar reason, neoadjuvant therapy was considered an exclusion criterion to reduce heterogeneity of the patient cohort and to remove additional unknown influence on sEV characteristics. Additionally, tumor sizes and histological grades were different. However, PDAC is inherently a heterogeneous disease, and the effectiveness of different regimes of systemic therapy differs among patients; from this point of view, an even larger group of patients would be necessary to better evaluate these effects on sEV. Nonetheless, all of our patients were operated in a single high-volume pancreatic surgery center following the same protocol, while biological samples were collected, stored and analyzed by dedicated researchers following a standard protocol. Despite known limitations of the NTA for EV analysis, the approach to plasma sEV quantification used here was previously supported by orthogonal techniques such as asymmetric flow field flow fractionation coupled to detectors and transmission electron microscopy [34,56].

## 5. Conclusions

To our knowledge, there are no studies in any cancer that evaluated the association between sEV characteristics and feasibility of tumor resection and its radicality. Our proof-of-principle study has managed to associate preoperatively acquired sEV characteristics with feasibility of resection and its radicality in PDAC patients undergoing surgery with curative intent. Furthermore, we were able to associate patients with worse OS with these preoperatively acquired sEV characteristics. Liquid biopsy with sEV provides new perspective in prognostic stratification of PDAC patients and, consequently, optimization of their treatment. However, further studies are necessary to support our observations and evaluate the importance of plasma sEV in clinical practice for PDAC.

## Figures and Tables

**Figure 1 cancers-15-00605-f001:**
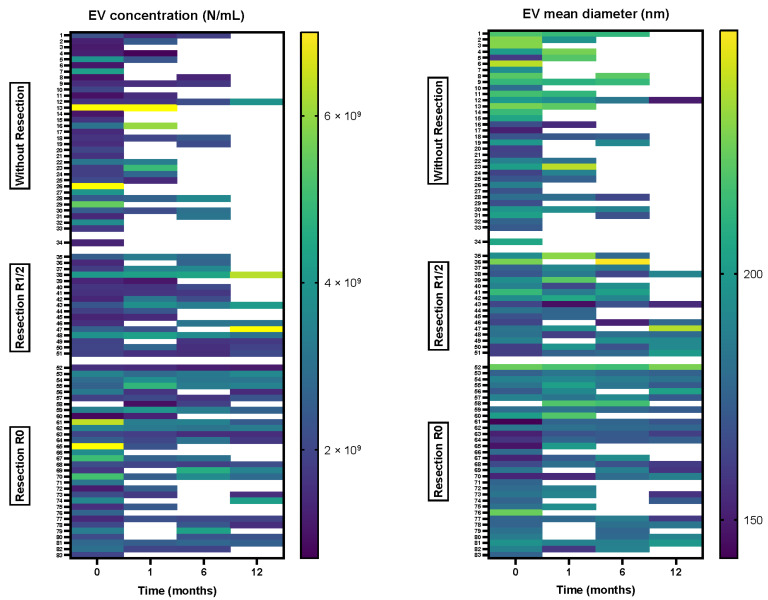
Heatmaps comparing sEV concentration (**left**) and mean size (**right**) for PDAC patients at different time points during follow-up. Patients were grouped according to type and radicality of surgery. Values outside the defined range are presented in yellow. Samples unavailable due to losing patients during follow-up are indicated in white.

**Figure 2 cancers-15-00605-f002:**
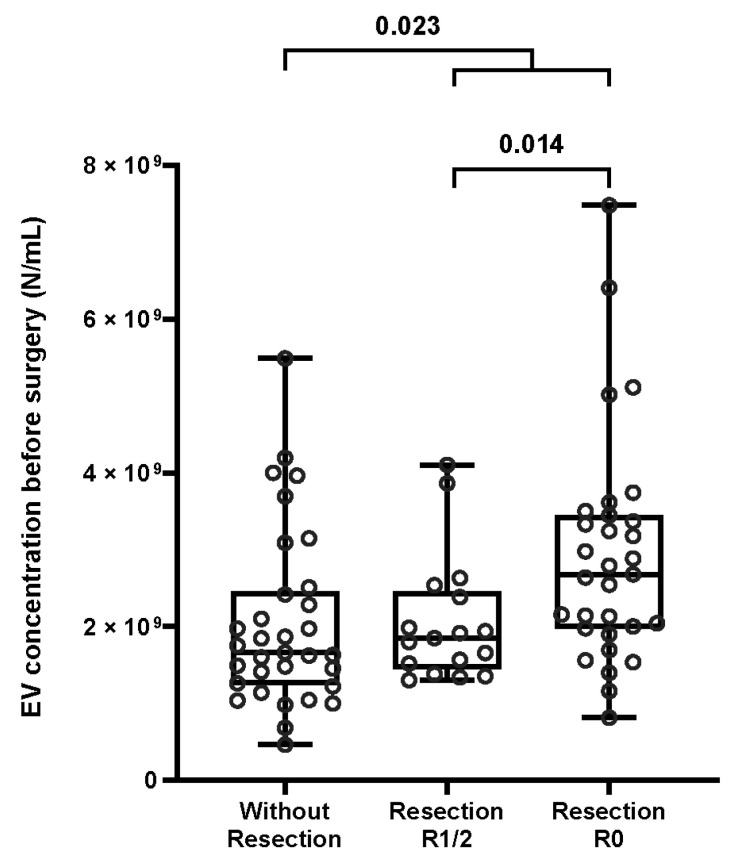
Box plot representing sEV concentrations before surgery in different subgroups of patients. Patients with resection had significantly higher concentrations of sEV than patients without resection (*p* = 0.023). Patients with R0 resection had significantly higher concentrations of sEV than patients with R1/R2 resection (*p* = 0.014).

**Figure 3 cancers-15-00605-f003:**
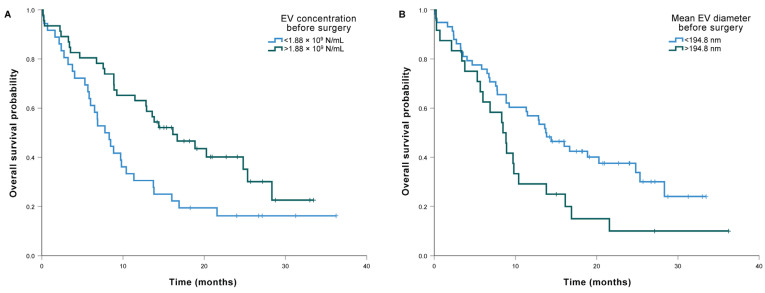
OS of patients with PDAC according to preoperative sEV characteristics. (**A**) Association of sEV concentration before surgery with OS. If sEV concentration was more than 1.88 × 10^9^/mL, patients had longer OS. (**B**) Association of sEV mean diameter with OS. If sEV mean diameter was less than 194.8 nm, patients had longer OS. Censored patients are presented with a vertical line.

**Table 1 cancers-15-00605-t001:** Baseline patients’ characteristics.

Variables		Study Patients N = 83	W/O Resection N = 33	With Resection N = 50	*p*-Value *
Sex	Male, N (%)	51 (61.4)	22 (66.7)	29 (58.0)	0.494 ^d^
	Female, N (%)	32 (38.6)	11 (33.3)	21 (42.0)	
Age	Years, median (25–75%)	70 (63–77)	71 (65–77.5)	69.5 (61.0–77.0)	0.536 ^e^
ASA score	1, N (%)	1 (1.2) {1}	0 (0.0)	1 (2.0) {1}	0.247 ^d^
	2, N (%)	21 (25.6)	6 (18.2)	15 (30.6)	
	3, N (%)	60 (73.2)	27 (81.8)	33 (67.3)	
Smoking	No, N (%)	34 (42.5) {3}	9 (29.0) {2}	25 (51.0) {1}	0.065 ^d^
	Yes, N (%)	46 (57.5)	22 (71.0)	24 (49.0)	
Alcohol consumption	Never, N (%)	19 (24.4) {5}	8 (26.7) {3}	11 (22.9) {2}	0.618 ^d^
	Occasional, N (%)	40 (51.3)	15 (50.0)	25 (52.1)	
	Moderate, N (%)	14 (17.9)	4 (13.3)	10 (20.8)	
	Heavy, N (%)	5 (6.4)	3 (10.0)	2 (4.2)	
BMI 6 months before surgery	kg/m^2^, median (25–75%)	26.4 (23.9–30.7) {4}	29.0 (25.7–31.4) {3}	25.7 (22.8–29.7) {1}	0.014 ^e^
BMI at surgery	kg/m^2^, median (25–75%)	24.9 (22.0–28.4) {4}	25.4 (23.8–28.4) {3}	23.8 (21.5–28.4) {1}	0.179 ^e^
WBC count ^a^	* 10^9^/L, median (25–75%)	7.5 (5.9–8.6) {3}	8.1 (5.95–9.15)	7.3 (5.9–8.4) {3}	0.200 ^e^
CRP ^a^	mg/L, median (25–75%)	5 (5–21.25) {3}	10 (5–38.5)	5 (5–9) {3}	0.002 ^e^
CA19-9 ^a^	median (25–75%)	1921 (329–10,457.5) {2}	4532 (354–21,598.25) {1}	884.0 (231.5–4189.0)	0.105 ^e^
CEA ^a^	median (25–75%)	3.0 (1.6–5.7) {2}	3.5 (1.8–8.1) {2}	2.5 (1.6–4.7)	0.145 ^e^
Preoperatively evaluated tumor size	mm, median (25–75%)	30 (25–41) {10}	33 (25–45) {1}	28 (24.5–36.5) {9}	0.169 ^e^
Distant metastases ^b^	No, N (%)	63 (75.9)	16 (48.5)	47 (94.0)	<0.001 ^d^
	Yes, N (%)	20 (24.1)	17 (51.5)	3 (6.0)	
Tumor differentiation ^c^	Poor, N (%)	39 (53.4) {10}	14 (56.0) {8}	25 (52.1) {2}	0.047 ^d^
	Moderate, N (%)	32 (43.8)	11 (44.0)	21 (43.8)	
	Well, N (%)	2 (2.7)	0 (0.0)	2 (4.2)	
Adjuvant chemotherapy †	No, N (%)	33 (40.7) {2}	17 (53.1) {1}	16 (32.7) {1}	0.105 ^d^
	Yes, N (%)	48 (59.3)	15 (46.9)	33 (67.3)	
Resection radicality ^c^	R0, N (%)			24 (49.0) {1}	
	R0 (<1 mm), N (%)			8 (16.3)	
	R1, N (%)			15 (30.6)	
	R2, N (%)			2 (4.1)	
pT stage	1, N (%)	1 (1.3) {3}	0 (0.0) {3}	1 (2.0)	<0.001 ^e^
	2, N (%)	16 (20.0)	0 (0.0)	16 (32.0)	
	3, N (%)	33 (41.3)	1 (3.3)	32 (64.0)	
	4, N (%)	30 (37.5)	29 (96.7)	1 (2.0)	
pN stage	0, N (%)	9 (18.0) {33}		9 (18.0) {33}	
	1, N (%)	29 (58.0)		29 (58.0)	
	2, N (%)	12 (24.0)		12 (24.0)	

w/o: without; ASA: American Association of Anesthesiologists; BMI: body mass index; WBC: white blood cell; CRP: C-reactive protein; CA 19-9: carbohydrate antigen 19-9; CEA: carcinoembryonic antigen; { }: number of missing data in each category. Data collected immediately before surgery (a), intraoperatively (b) or by definite histology (c). † Adjuvant chemotherapy for all 48 patients who received it was initiated more than one month after surgery. Fourteen of those additionally received radiation therapy more than one month after surgery. * Comparison between patients w/o resection and patients with resection was calculated using Fisher’s exact test (d) or Mann–Whitney test (e).

**Table 2 cancers-15-00605-t002:** Patients’ small plasma extracellular vesicle (EV) characteristics.

		Study Patients	W/O Resection	With Resection	
	Small EV Characteristics	Median (25–75%)	Median (25–75%)	Median (25–75%)	*p*-Value ^#^
Before surgery (N = 82: 33 w/o resection, 49 with resection)	Concentration (N × 10^9^/mL)	1.97 (1.49–3.10)	1.66 (1.24–2.46)	2.14 (1.61–3.29)	0.023
	Mean diameter (nm)	182.6 (170.1–199)	195.2 (169.9–213.8)	181.4 (170–189.1)	0.057
After one month (N = 53: 16 w/o resection, 37 with resection)	Concentration (N × 10^9^/mL)	2.29 (1.61–3.22)	2.08 (1.46–2.71)	2.38 (1.71–3.28)	0.333
	Mean diameter (nm)	188.9 (176.5–204.2)	196.2 (180.5–218.9)	185.7 (176.4–196.4)	0.075
After six months (N = 43: 9 w/o resection, 34 with resection)	Concentration (N × 10^9^/mL)	2.68 (1.68–3.30)	2.15 (1.75–3.06)	2.78 (1.88–3.32)	0.385
	Mean diameter (nm)	183.8 (176.6–192.6)	188.5 (171.1–213)	181.2 (177.2–191.2)	0.471
After 12 months (N = 29: 1 w/o resection, 28 with resection)	Concentration (N × 10^9^/mL)	2.58 (1.84–3.47)	3.90	2.54 (1.79–3.42)	*
	Mean diameter (nm)	177.2 (166–191.3)	149.7	177.4 (170.7–191.9)	*

w/o: without; *: After 12 months, data for a single patient in the group without resection was available; thus, comparison between the two groups was not possible. # Comparison between patients w/o resection and patients with resection was calculated using Mann–Whitney test.

**Table 3 cancers-15-00605-t003:** Association of sEV characteristics before surgery with OS.

	Small EV Characteristics	HR (95% CI) *	*p*-Value	HR (95% CI) _adj_ *	*p*-Value _adj_
Study Patients	Concentration (N × 10^9^/mL)	1.00 (1.00–1.00)	0.069	1.00 (1.00–1.00)	0.599
	Mean diameter (nm)	1.10 (0.98–1.24)	0.121	0.92 (0.80–1.05)	0.224
w/o Resection	Concentration (N × 10^9^/mL)	1.00 (1.00–1.00)	0.895	1.00 (1.00–1.00)	0.714
	Mean diameter (nm)	0.81 (0.68–0.97)	0.021	0.85 (0.71–1.01)	0.065
With Resection	Concentration (N × 10^9^/mL)	1.00 (1.00–1.00)	0.407	1.00 (1.00–1.00)	0.535
	Mean diameter (nm)	1.10 (0.89–1.34)	0.381	1.07 (0.86–1.32)	0.555

Adj: adjustment for clinical parameters: study patients—adjustment for CA19-9, type of surgery and adjuvant chemotherapy; patients without resection—adjustment for CRP and distant metastases; patients with resection—adjustment for CA19-9. * HR and 95% CI are reported for a difference of 10 units.

**Table 4 cancers-15-00605-t004:** Association of sEV characteristics before surgery with OS after surgery using stratification based on ROC curve analysis.

Small EV Characteristics		Survival Months, Median (25–75%)	HR (95% CI)	*p*-Value	HR (95% CI) _adj_	*p*-Value _adj_
Concentration (N × 10^9^/mL)	<1.88 × 10^9^/mL	7.8 (3.8–13.8)	Reference		Reference	
	>1.88 × 10^9^/mL	16.1 (7.7–28.3)	0.54 (0.32–0.90)	0.018	0.74 (0.41–1.35)	0.325
Mean diameter (nm)	<194.8 nm	13.7 (6.5–28.3)	Reference		Reference	
	>194.8 nm	8.5 (3.8–13.8)	1.81 (1.06–3.10)	0.030	0.56 (0.28–1.12)	0.100

Adj: adjustment for CA19-9, type of surgery and adjuvant chemotherapy.

## Data Availability

The data presented in this study are available on request from the corresponding authors. The data are not publicly available due to ethical reasons.

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
