# Peer review of "Plasma Extracellular Vesicle Characteristics as Biomarkers of Resectability and Radicality of Surgical Resection in Pancreatic Cancer—A Prospective Cohort Study"

_cancers, 2023, doi:10.3390/cancers15030605_

Round 1

Reviewer 1 Report

The University of California research team published a paper in the journal Communications Medicine entitled "Early-stage multi-cancer detection using anextracellular vesicle The protein-based blood test (EVs), which detects Extracellular Vesicles (EVS) in blood, is expected to provide a new method for the early screening and diagnosis of pancreatic cancer.

The significance of EVs in the diagnosis and prognosis of pancreatic cancer is worth further exploration in clinic. In this paper, a prospective clinical study was designed to confirm the predictive value of EVs in resectomies and prognosis of pancreatic cancer, which is innovative to a certain extent, but the design has the following problems:

1. The proportion of patients with distant metastases at baseline, how did the authors evaluate these patients as resectable patients at the enrollment stage?

2. The author did not conduct statistical or stratified analysis on the T and N stages of patients, which have independent predictive value for the surgical outcome of patients.

3. The author regards neoadjuvant therapy as the exclusion criteria, but in fact, neoadjuvant therapy plays an important role in perioperative treatment of pancreatic cancer at present. Perhaps the inclusion of neoadjuvant patients and monitoring of EVs concentration before and after surgery are better research directions.

In short, there are many confounding factors in this study, and no detailed stratification has been obtained. The conclusions are too general and rough.

Author Response

REVIEWER #1

The University of California research team published a paper in the journal Communications Medicine entitled "Early-stage multi-cancer detection using an extracellular vesicle The protein-based blood test (EVs), which detects Extracellular Vesicles (EVS) in blood, is expected to provide a new method for the early screening and diagnosis of pancreatic cancer.

A: We thank the reviewer to referring us to the study we were not aware of and is certainly very interesting and worth reading. In the future, early recognition of PDAC is surely going to be one of the most important points studied in order to improve patients’ survival. Due to its relevance to our study, we have included and cited the proposed paper in our manuscript (ref. 17; page 2, rows 90,91; page 3, rows 103-105). Nonetheless, this article focuses on screening and establishing the (early) diagnosis of pancreatic cancer, while the aim of our study was to predict feasibility and radicality of resection in patients already diagnosed with PDAC as discussed below.

Q1: The proportion of patients with distant metastases at baseline, how did the authors evaluate these patients as resectable patients at the enrollment stage?

A1: Thank you for the comment. All of the patients enrolled in our study were presumed to have a resectable PDAC based on preoperative diagnostic workup; they all had a CT scan performed and were presented at a multidisciplinary team meeting that also indicated surgery; none of the enrolled patients were suspected to have distant metastases or locally unresectable disease before the operation. However, as also evaluated in the Cochrane Database Systematic Review from 2013 (and cited in our manuscript: page 2, rows 59-61), even modern radiological workup may misjudge the actual resectability status of PDAC, as up to 23% of presumably resectable tumors are found unresectable or metastatic at surgery. Furthermore, also in discussion (page 11, rows 349,350) we have pointed out that CT scan has limited sensitivity for discovering small liver or peritoneal metastatic lesions (ref. 43)

In our study group without resection (Table 1), there were 17 patients found to have distant metastases at surgical exploration and in these cases resection was not attempted. Among the patients with resection, there were three to have distant metastases which were confirmed only at the final histology report: the first patient was found to have a focus of peritoneal carcinomatosis on the resected specimen of the pancreas, in the second patient the final histology report confirmed a small solitary liver metastasis (which was also sent to frozen section during surgery but the pathologist was then unable to confirm its malignancy) and the third patient had a large tumor demanding a multivisceral resection (distal pancreatectomy, splenectomy, partial gastrectomy and splenic flexure resection), where small lesions were found in the specimen.

Preoperatively known distant metastases were an exclusion criterion for our study and, thus, such patients were never candidates for enrollment. We have included a more detailed enrollment process description also in the manuscript (page 3, rows 123-126).

Q2: The author did not conduct statistical or stratified analysis on the T and N stages of patients, which have independent predictive value for the surgical outcome of patients.

A2: Thank you for your comment. We included data on pathological (p) T and N stage in Table 1; previous variable “Positive lymph nodes” was deleted, because it is now included in the “pN stage” variable. Significantly higher pT stage was seen in patients with resection, which we included in the text of the manuscript (page 5, rows 214,215).

We also added the statistical analysis of the association of pT and pN stages with overall survival. Higher pT stage was significantly associated with shorter survival in univariable analysis, which we included in the text of the manuscript (page 9, rows 301-303). On the other hand, no significant association was observed in multivariable analysis. Unfortunately, N stage was available only for patients with resection, limiting its predictive value in our study group. 

Table: Univariable analysis of association of pT and pN stages with OS 

HR (95% CI) 

P 

pT 

1+2 

Reference 

2.33 (0.87-6.24) 

0.091 

9.02 (3.43-23.71) 

<0.001 

pN* 

Reference 

1.13 (0.38-3.39) 

0.828 

1.58 (0.44-5.64) 

0.482 

*available only for patients with resection 

Q3: The author regards neoadjuvant therapy as the exclusion criteria, but in fact, neoadjuvant therapy plays an important role in perioperative treatment of pancreatic cancer at present. Perhaps the inclusion of neoadjuvant patients and monitoring of EVs concentration before and after surgery are better research directions.

A3: The current European Society for Medical Oncology (ESMO) and National Comprehensive Cancer Network (NCCN) guidelines (ref. 4 and 40) still propose upfront resection in resectable PDAC, thus surgery remains the gold standard of treatment. Neoadjuvant therapy has been utilized in borderline resectable PDAC in order to downstage the disease or provide better chances of negative resection margins, and such has also been the regime in our hospital. However, we did not want to include these patients in our study group as this would bring even more heterogeneity to it, especially as systemic treatment could possibly alter the characteristic of plasma EVs (as mentioned in discussion: page 12, rows 417-419). We have now included this additional comment also for neoadjuvant therapy to avoid potential further misunderstandings. Nonetheless, we absolutely agree that neoadjuvant treatment is becoming more and more recognizable and established in treatment of even resectable PDAC. Yet, this is probably going to become the standard of care in the future and our study also aims to support this; based on EVs, we aimed to preoperatively presume feasibility of PDAC resection and also suggest clinical benefit of recognizing patients with high chances of positive resection margins in order to back up neoadjuvant treatment for them. This has additionally been noted in Discussion (page 12, rows 400-408).

As suggested by the reviewer, a study including patients with neoadjuvant therapy and monitoring EVs characteristics and thus PDAC response to treatment is absolutely necessary in the future. However, as our national and European guidelines suggest neoadjuvant treatment only in borderline cases, it is highly unlikely to enroll all (also resectable) PDAC patients to such treatment regime. But once there is enough data to push neoadjuvant therapy for resectable disease forward also in Europe (as has been the case in the USA), there will be an opportunity for us to extend our cohort to such extent.

Reviewer 2 Report

The manuscript "Plasma extracellular vesicle characteristics as biomarkers of resectability and radicality of surgical resection in pancreatic cancer – a prospective cohort study" is an interesting contribution to the field. The author has examined sEV’s prospective role as a biomarker with respect to biomarkers of tumor resectability, radicality of resection, and overall survival in pancreatic ductal adenocarcinoma (PDAC) patients. The number of PDAC patients included in this study was 83. The authors purified sEVs from 1 mL of plasma. Samples were handled to avoid damaging EVs. The isolated sEVs were quantified for size and concentration using NTA. While contamination of isolated sEVs was checked using ELISA for ApoA1. The author showed higher sEV concentration in preoperative patients undergoing resection than those without resection. They also showed significant sEV concentration change between R0 vs R1/R2. The author portrayed sEV size and concentration as a biomarker between preoperative PDAC and its feasibility and radicality of PDAC resection that also enable discrimination of patients with worse overall survival. The manuscript is thorough and well-written with clear graphical representations The present study is an appreciable attempt in the field of biomarkers with respect to circulatory EVs in PDAC. Although the objective and approach are appropriate, the authors do not succeed in presenting the obtained results in a fully understandable and convincing way. The manuscript can be improved by addressing the following suggestions:

1)    I find that NTA is not enough to support these assumptions/claims. If the author intends to use the size and concentration of sEVs as a biomarker in PDAC. It’s suggested to add two other techniques for size and concentrations of sEVs so that the obtained results can be compared, and results can be concluded accordingly. Kindly, add results obtained from either “Zetasizer/DLS or NanoFCM or izon exoid” too for the characteristics of isolated sEVs. It’s highly suggested to add results from two of these techniques given author’s claim is entirely based on sEV characteristics.

2)     No doubt the author has used all the pre-proven and published techniques for the isolation and purification of the EVs in the study. Yet as per guidance by MISEV 2018 for studies focusing on EVs, there should be a quality check for the set of EVs isolated even when minor modifications are introduced in the published protocol. At a minimum, 3 different techniques for their size, integrity, and characterization. Given NTA is provided in. the draft and assuming “Zetasizer/DLS and NanoFCM” results will be added in the revised draft, an additional immunoblot/ELISA for the major sEV-associated proteins like CD9, CD63, CD81, ALIX, flotillin, and HSP70 is suggested to be added to the draft. If the author wishes they can opt to mix and match of immunoblot and ELISA for the aforementioned sEV-associated proteins.

3)    As per the guidelines of ISEV; all researchers are strongly encouraged to submit their experimental protocols on EV isolation and characterization to the EV-TRACK website (evtrack.org). The database helps to calculate the metric value for isolated EVs. When the author receives an EV-TRACK ID; kindly add it to the methods in your draft.

4)    For NTA analysis – author’s have mentioned camera level 14; could you please add “the number of particles selected per frame as well as the detection threshold?” For NTA analysis, all these factors are important, or else it will automatically lead to misleading and incomparable EV particles for final experiments. 

5)    It would be useful to describe (briefly) the exact context of use (COU) that this putative biomarker may serve for this population.

6)    The conclusion should be toned down.

Author Response

REVIEWER #2

The manuscript "Plasma extracellular vesicle characteristics as biomarkers of resectability and radicality of surgical resection in pancreatic cancer – a prospective cohort study" is an interesting contribution to the field. … The manuscript is thorough and well-written with clear graphical representations The present study is an appreciable attempt in the field of biomarkers with respect to circulatory EVs in PDAC. Although the objective and approach are appropriate, the authors do not succeed in presenting the obtained results in a fully understandable and convincing way.

A: Thank you for your positive comments of our work. We have tried to address all remaining issues in our responses to reviewer’s questions below.

Q1: I find that NTA is not enough to support these assumptions/claims. If the author intends to use the size and concentration of sEVs as a biomarker in PDAC. It’s suggested to add two other techniques for size and concentrations of sEVs so that the obtained results can be compared, and results can be concluded accordingly. Kindly, add results obtained from either “Zetasizer/DLS or NanoFCM or izon exoid” too for the characteristics of isolated sEVs. It’s highly suggested to add results from two of these techniques given author’s claim is entirely based on sEV characteristics.

A1: Thank you for this relevant comment. Ideally, any measured EV characteristic should be confirmed by an orthogonal method, but this is a lot easier done in the “in vitro” (fundamental) studies then in the clinical biomarkers studies. Clinical studies are namely limited by large sample numbers (here we processed 207 samples from 82 patients) and low biofluid volume (1 mL). To improve confidence in our approach to plasma EV quantification, we have thus performed a separate extensive study on a smaller plasma sample number published in Scientific Reports - Nature (Holcar et al., 2020, EV-TRACK ID: EV200196), where NTA quantification data was supported by the AF4-MALS-UV and TEM data. We have since successfully used our approach in several published studies (Štok et.al., Cells, 2020; Levstek et. al, Genes, 2021; Badovinac, Journal of Personalised Medicine, 2021). In our so-far unpublished work on 208 individual plasma samples, we have additionally shown that (i) using sUC sEV enrichment approach (as here), we successfully remove most of the lipoproteins that could affect NTA measurements, as only 0.0056% of ApoA1, 0.00173% of ApoB100 and below the detection limit of ApoB48 remain in the sEV sample; (ii) NTA quantification after sUC enrichment of EVs from human plasma is repeatable, as coefficient of variation (CV) of sEV concentration is 13.27 % and CV of „sEV“ mean diameter is 4.10 %. We would also like to add here, that we are using an automated sample assistant for sample injection into NanoSight NS300 to limit operators’ effect on the measurements. In addition to quantification of plasma sEVs, we have used NTA to quantify urinary EVs and have shown that NTA data correlated with the TEM data (Sedej et al., JEV, 2022). All of the additionally mentioned research has now been also cited in the manuscript (ref. 36,37,56). Still, to remind the reader of this limitation we have added text to the Discussion section (page 13, rows 434-438).

Q2: No doubt the author has used all the pre-proven and published techniques for the isolation and purification of the EVs in the study. Yet as per guidance by MISEV 2018 for studies focusing on EVs, there should be a quality check for the set of EVs isolated even when minor modifications are introduced in the published protocol. At a minimum, 3 different techniques for their size, integrity, and characterization. Given NTA is provided in. the draft and assuming “Zetasizer/DLS and NanoFCM” results will be added in the revised draft, an additional immunoblot/ELISA for the major sEV-associated proteins like CD9, CD63, CD81, ALIX, flotillin, and HSP70 is suggested to be added to the draft. If the author wishes they can opt to mix and match of immunoblot and ELISA for the aforementioned sEV-associated proteins.

A2: Thank you for bringing MISEV guidelines to the attention! In the ISEV Science & Meetings committee we are happy to see MISEV guidelines implemented in the review process to improve the quality of the EV studies. Still, we are also trying to spread awareness that MISEV guidelines are not a blueprint or a checklist of “dos and don’ts”, and that common sense relevant to the studied system should be applied (this is specifically stated in the new guidelines coming out soon). In the here described study, we have used exactly the same protocol for EV enrichment from plasma extensively studied in Holcar et al (Scientific Reports, 2020), uploaded into EV-TRACK (ID EV200196), and used in several other published studies (Badovinac, Journal of Personalised Medicine, 2021; Štok et.al., Cells, 2020; Levstek et. al, Genes, 2021). We believe this sufficiently supports our approach to EV enrichment from plasma. To further stress the support for here used EV enrichment method, we have included mentioned references to the Methods section (page 4, rows 161-163).

Q3: As per the guidelines of ISEV; all researchers are strongly encouraged to submit their experimental protocols on EV isolation and characterization to the EV-TRACK website (evtrack.org). The database helps to calculate the metric value for isolated EVs. When the author receives an EV-TRACK ID; kindly add it to the methods in your draft.

A3: Thank you for the reminder. We have added the EV-TRACK ID of the here used protocol (EV200196) to the Methods section (page 4, rows 161-163).

Q4: For NTA analysis – author’s have mentioned camera level 14; could you please add “the number of particles selected per frame as well as the detection threshold?” For NTA analysis, all these factors are important, or else it will automatically lead to misleading and incomparable EV particles for final experiments.

A4: Thank you for the comment. As noticed by the reviewer, the number of particles per frame (PPF) is indicative of the quality of NTA measurement, with optimal PPF range for concentration determination between 10 and 100. To further limit any putative influence of sEV sample concentration on NTA EV quantification, we have diluted all sEV samples in the range of 20-30 PPF for the NTA measurement (new Table S1 in Suppl. material). For the detection threshold we use 5 in our all measurements. We have added a more detailed description of the NTA measurement in the Methods section (page 4, rows 167-177).

Table S1: Particles per frame for NTA measurements of plasma sEV samples

Study patients

w/o resection

With resection

Median PPF

(25-75%)

Median PPF

(25-75%)

Median PPF

(25-75%)

Before surgery (N=82: 33 w/o resection, 49 with resection)

20.48

(15.29-26.93)

18.65

(14.65-24.80)

22.15

(16.00-27.50)

After one month (N=53: 16 w/o resection, 37 with resection)

23.75

(16.45-33.35)

21.23

(15.70-33.54)

25.70

(18.20-32.70)

After six months (N=43: 9 w/o resection, 34 with resection) 35

24.70

(19.35-32.65)

20.70

(16.00-24.00)

25.25

(19.80-33.60)

After 12 months (N=29: 1 w/o resection, 28 with resection)

21.70

(17.65-31.15)

32 *

21.65

(17.65-29.80)

PPF: particles per frame; *: After 12 months, data for a single patient in the group without resection was available

Q5: It would be useful to describe (briefly) the exact context of use (COU) that this putative biomarker may serve for this population.

A5: Thank you for the remark about the necessity to summarize our findings. Based on EV characteristics obtained even before surgery, we could presume which patients would be able to have their PDAC removed and further, which of these would be able to have PDAC removed radically (with negative resection margins). Even in PDAC deemed resectable on the established preoperative diagnostic workup (CT, EUS), it is impossible to predict actual feasibility and radicality of resection, as both these features are often directly connected to tumor biology and microscopic environment. Yet, such preoperatively acquired data could assist in decision to postpone surgery in PDAC for neoadjuvant therapy in order to dowstage the tumor (and provide higher rates of radical resection after neoadjuvant treatment) or avoid surgery with possible complications altogether (in those patients with unfavorable tumor biology).
Rationale for such research are discussed in detail in Introduction (first, second and fifth paragraph) and benefits of our findings are now more clearly summarized in Discussion (page 12, rows 397-408) and in Conclusions.

Q6: The conclusion should be toned down.

According to your proposal, we have toned down our conclusions and have stressed that our research was a proof-of-principle study and that it should be evaluated with another independent cohort study (page 3, row 110; pages 12,13, rows 397-448). Certain modifications have also been made throughout Discussion.

Round 2

Reviewer 1 Report

At present, the design method and discussion of the article have been greatly improved, and the language needs to be further polished to further improve the readability of the article

Reviewer 2 Report

The revised MS draft is ready to be accepted.